# Single-Cell Transcriptomic Profiling Identifies Molecular Phenotypes of Newborn Human Lung Cells

**DOI:** 10.3390/genes15030298

**Published:** 2024-02-26

**Authors:** Soumyaroop Bhattacharya, Jacquelyn A. Myers, Cameron Baker, Minzhe Guo, Soula Danopoulos, Jason R. Myers, Gautam Bandyopadhyay, Stephen T. Romas, Heidie L. Huyck, Ravi S. Misra, Jennifer Dutra, Jeanne Holden-Wiltse, Andrew N. McDavid, John M. Ashton, Denise Al Alam, S. Steven Potter, Jeffrey A. Whitsett, Yan Xu, Gloria S. Pryhuber, Thomas J. Mariani

**Affiliations:** 1Department of Pediatrics, University of Rochester Medical Center, Rochester, NY 14642, USA; gautam_bandyopadhyay@urmc.rochester.edu (G.B.); stephen_romas@urmc.rochester.edu (S.T.R.); heidie_huyck@urmc.rochester.edu (H.L.H.); ravi_misra@urmc.rochester.edu (R.S.M.); gloria_pryhuber@urmc.rochester.edu (G.S.P.); tom_mariani@urmc.rochester.edu (T.J.M.); 2Genomic Research Center, University of Rochester Medical Center, Rochester, NY 14642, USA; jacquelyn.myers@stjude.org (J.A.M.); cameron_baker@urmc.rochester.edu (C.B.); jason.myers@stjude.org (J.R.M.); john_ashton@urmc.rochester.edu (J.M.A.); 3Department of Pediatrics, Cincinnati Children’s Hospital Medical Center, Cincinnati, OH 45219, USA; minzhe.guo@cchmc.org (M.G.); steve.potter@cchmc.org (S.S.P.); jeffrey.whitsett@cchmc.org (J.A.W.); yan.xu@cchmc.org (Y.X.); 4Lundquist Institute for Biomedical Innovation, Harbor-UCLA Medical Center, University of California Los Angeles, Los Angeles, CA 90024, USA; soula.danopoulos@lundquist.org (S.D.);; 5Clinical & Translational Science Institute, University of Rochester, Rochester, NY 14642, USA; jennifer_dutra@urmc.rochester.edu (J.D.); jeanne_wiltse@urmc.rochester.edu (J.H.-W.); 6Department of Biostatistics and Computational Biology, University of Rochester Medical Center, Rochester, NY 14642, USA; andrew_mcdavid@urmc.rochester.edu

**Keywords:** single-cell RNAseq, matrix fibroblast, lung development, newborn lung

## Abstract

While animal model studies have extensively defined the mechanisms controlling cell diversity in the developing mammalian lung, there exists a significant knowledge gap with regards to late-stage human lung development. The NHLBI Molecular Atlas of Lung Development Program (LungMAP) seeks to fill this gap by creating a structural, cellular and molecular atlas of the human and mouse lung. Transcriptomic profiling at the single-cell level created a cellular atlas of newborn human lungs. Frozen single-cell isolates obtained from two newborn human lungs from the LungMAP Human Tissue Core Biorepository, were captured, and library preparation was completed on the Chromium 10X system. Data was analyzed in Seurat, and cellular annotation was performed using the ToppGene functional analysis tool. Transcriptional interrogation of 5500 newborn human lung cells identified distinct clusters representing multiple populations of epithelial, endothelial, fibroblasts, pericytes, smooth muscle, immune cells and their gene signatures. Computational integration of data from newborn human cells and with 32,000 cells from postnatal days 1 through 10 mouse lungs generated by the LungMAP Cincinnati Research Center facilitated the identification of distinct cellular lineages among all the major cell types. Integration of the newborn human and mouse cellular transcriptomes also demonstrated cell type-specific differences in maturation states of newborn human lung cells. Specifically, newborn human lung matrix fibroblasts could be separated into those representative of younger cells (*n* = 393), or older cells (*n* = 158). Cells with each molecular profile were spatially resolved within newborn human lung tissue. This is the first comprehensive molecular map of the cellular landscape of neonatal human lung, including biomarkers for cells at distinct states of maturity.

## 1. Introduction

The lung is a complex organ comprised of over 40 different cell types [1,2]. Despite recent advances in our understanding of lung development, the complex cellular function and intercellular interactions in the developing human lung are yet to be clearly understood. Development and maintenance of lung structure requires cross talk among multiple cell types to coordinate lineage specification, cell proliferation, differentiation, migration, morphogenesis, and injury repair. The diverse array of pulmonary cells can be categorized into four major cell populations, namely, epithelial cells, endothelial cells, mesenchymal cells, and lung-resident and -transient immune cells, with each group being relatively well-distinguished by specific cell-surface proteins. Even though key signaling molecules, genes, and pathways driving lung development have been identified [3,4,5,6,7,8,9], significant knowledge gaps still exist in our understanding of this process especially in humans and interdisciplinary efforts will be required to further our understanding of the early life origin of lung disease [10].

The transition of the lung from fetal to neonatal states is highly complex, and has been characterized in murine models [11,12,13,14,15]. Cell lineages, and their relationships, during lung development and in diseased states have been extensively studied in rodent models, thanks in large part to use of transgenic technology. It has been more difficult to confirm independent cell types and their lineages in the relatively rare and non-experimental nature of human lung tissue analysis. While early stages of human fetal lung development have been characterized at the molecular level [16,17], data describing the newborn human lung is lacking [18]. Although molecular profiling has been applied to pre-viable human lung [17,19], further understanding of later human lung development has been limited by lack of access to tissue of sufficient quality for molecular analysis. These limitations have been recently overcome by the NHLBI Molecular Atlas of Lung Development Program (LungMAP). The establishment of the LungMAP program, and its success in obtaining human tissues for structural, cellular and molecular analysis has, and will continue to, facilitate this progress [20,21,22,23]. The LungMAP consortium was formed to create a molecular atlas of the developing human lung that incorporates detailed structural and molecular mechanisms involved normal perinatal and neonatal lung development. The LungMAP website (https://www.LungMAP.net; accessed on 1 February 2024) serves as a repository of datasets generated from multiple species across multiple omics platforms. The portal also incorporates novel computational tools for the analysis and interpretation of omics and image data that have been extensively used by researchers in the pulmonary field [24]. Additionally, most high-throughput molecular studies of lung development have used whole lung tissue [11], limiting insights into the activities of and interactions among different cell types. However, given recent advancements in high-throughput molecular profiling technologies, rapid progress is being made [25,26,27]. Single-cell RNA-seq enables transcriptomic mapping of individual cells, to measure and understand cellular heterogeneity and mechanistic responses in complex biological systems [27]. It offers an opportunity to explore the whole transcriptome of individual cells, and thereby organizing the cells into cellular states in an unsupervised manner. Single-cell RNA-seq can be performed in a multitude of organisms, which allows for unbiased comparisons across different species [28]. Recently, the Human Cell Atlas consortium has prioritized the characterization of cellular composition of lung tissue using single-cell sequencing [29]. Single-cell RNA-Seq has also been applied for the identification of known or novel cell populations and to assess cellular heterogeneity and gene expression in changes in lung cell populations during health and diseases [20,30,31,32,33,34,35,36,37]. Here, we report transcriptional analysis of donor newborn human lung cells, including all major cell types, and describe the molecular profile for matrix fibroblast subtypes that may represent cells at different stages of development. 

LungMAP was developed to generate detailed structural and molecular data regarding normal perinatal and postnatal lung development in mice and humans [21,38]. We have recently reported that high-throughput analysis (transcriptomics, proteomics, etc.) of sorted dissociated cells from human neonatal and pediatric lungs reveals retention of in vivo phenotypes [19,21,39,40]. Here, we build on the rapid advancement in single-cell transcriptomics that enables the identification of cell-type specific transcriptomes of neonatal and ageing murine lungs, serving as a comparative basis for understanding the transcriptomic landscape of the newborn human lung [25,27]. We computationally integrated single-cell signatures of donor newborn human lungs with single-cell transcriptomic profiles of developing perinatal mouse lung to generate a trans-species cellular impression of the developing lung.

## 2. Materials and Methods

### 2.1. Study Population

Two newborn (one-day old) lungs were donated for research and provided, with de-identified clinical data, through the federal United Network of Organ Sharing via the National Disease Research Interchange (NDRI) and International Institute for Advancement of Medicine (IIAM). While both the lungs were from individuals who were deceased at one day of life due to diagnosed anencephaly, they differed at their gestational age at birth (GAB). While both the donors were female, one of them (Donor 1) was a full term (GAB of 38 weeks), whereas the other (Donor 2) had a pre-term birth (GAB of 31 weeks). The organs were received by the LungMAP Human Tissue Core at the University of Rochester, and subjected to processing as previously described [39]. The LungMAP program and resulting studies are approved by the University of Rochester IRB (RSRB00047606). Sample demographic information for the donor lungs used in the study have been presented in Appendix A.

### 2.2. Single-Cell Suspension Preparation

The right upper and middle lung lobes were digested to single-cell suspensions using a four-enzyme cocktail (collagenase A, DNase, dispase and elastase) according to LungMAP protocol, as described previously [39]. Isolated cells were resuspended in freezing media (90% FBS, 10% DMSO) at a concentration no more than 60 × 10^6^ cells/mL, slow cooled to −80 °C overnight and stored in liquid nitrogen until use. 

### 2.3. Human Single-Cell Sequencing 

Unfractionated dissociated cells from each subject were rapidly thawed and, without resting, used for two separate captures of single cells for RNAseq. For each lung, one capture was preceded by removal of dead or dying cells through magnetic selection using a Dead Cell Removal Kit (Miltenyi Biotech, Santa Barbara, CA, USA). Cell capture and library production was performed on the Chromium 10X Genomics system (v2 chemistry). Sequencing was performed on a HiSeq4000, with read alignment to GRCh38. Cells filtered to exclude low quality cells, and potential doublets, were used to create analytical dataset. Highly variable genes were identified using “FindVariableGenes” function in Seurat [41]. Principal component analysis (PCA) was used for dimension reduction based on only the highly variable genes. Top principal components (PCs) identified by JackStrawPlot() and graph-based Louvain-Jaccard methods [42] were used for t-Distributed Stochastic Neighbor Embedding (tSNE) and clustering analysis. All single-cell sequencing data analysis was performed using Seurat v2.4 [43,44]. Cluster markers were defined by differential expression using a parametric Wilcoxon rank sum test at a corrected significance level of *p* < 0.05 and upregulated the cluster compared to all other cells using the FindAllMarkers() command implemented in Seurat. Pathway analysis and cell type association was performed using ToppGene Functional Annotation tool (ToppFun) [45]. The human cell sequencing counts data is available for visualization and downloading through the LungMAP portal (https://lungmap.net/breath-omics-experiment-page/?experimentTypeId=LMXT0000000016&experimentId=LMEX0000004390&analysisId=LMAN0000000347&view=entitySet; accessed on 1 February 2024).

### 2.4. Mouse Single-Cell Sequencing 

Animal protocols were approved by the Institutional Animal Care and Use Committee at Cincinnati Children’s Medical Center (CCMC) in accordance with NIH guidelines. RNA from cells isolated from C57BL6/J mice (*n* = 2, at each time point) were used in the production of the mouse single-cell RNA-seq data set using Drop-Seq platform, as previously described [27]. Filtered data were log transformed, scaled, clustered and represented by t-Distributed Stochastic Neighbor Embedding (t-SNE), similar to the analysis of human cells. Cell clusters were assigned to putative cell types based on inspecting the expression of known cell type markers, and the individual cluster markers analyzed using ToppFun. Single-cell gene expression data from mouse lungs from all four time points have been made available on the Lung Gene Expression in Single Cell (LungGENS) web portal [19]. 

### 2.5. Integrating Human and Mouse Data 

We integrated our newborn human lung data set with single-cell sequencing data (Drop-Seq) from longitudinal postnatal (post-natal days 1, 3, 7, and 10) mouse lung samples [27] (Appendix A). These data characterized mouse datasets, hosted by LungMAP (https://lungmap.net; accessed on 1 February 2024), were used as a reference to help define the human lung cell populations. The useMart and getLDS functions, within the biomaRt package, were used to identify human and mouse orthologues (https://bioconductor.org/packages/release/bioc/html/biomaRt.html accessed on 1 February 2024). A list of 21,608 mouse MGI symbols were queried against HGNC symbols to identify 14,647 non-redundant orthologues. For all downstream analyses, HGNC symbols replaced MGI symbols within the mouse data set based on the biomaRt query [46]. We integrated human cells (*n* = 5499; 15%) and mouse cells (*n* = 32,849) using canonical correlation analysis (CCA) implemented within Seurat [43] (https://satijalab.org/seurat/ accessed on 1 February 2024) (Appendix A). No batch effects were evident following implementation of CCA. Marker genes from individual clusters were used to determine the cellular identity of co-clustered human cells in ToppFun.

### 2.6. Flow Cytometry 

The presence of immune cell populations in newborn human lungs were validated by flow cytometry, essentially as previously published [47]. Frozen lung cells were thawed, blocked (2% serum in 1% BSA/DPBS) and stained for anti-hCD45 (APC-R700, clone HI30), anti-hCD235a (PE-Cy5.5, clone GA-R2), anti-hCD3 (PE-Cy7, clone SK7) (all from BD Biosciences, San Jose, CA, USA) and anti-hHLA-DR (BV785, clone L243, Biolegend, San Diego, CA, USA) and 7-AAD (viability marker, BD Biosciences). Staining was assessed on a four-laser 18-color FACSAria flow cytometer (Becton Dickenson, San Jose, CA, USA). Single antibody stained Simply Cellular^®^ compensation beads (Bangs Lab, Fishers, IN, USA) were used for fluorescence overlap compensation. Fluorescence minus one (FMO) controls and heat-killed 7AAD stained cells were used to set expression gates for each antibody and for live/dead gating. Data were analyzed using FlowJo software (version 10; FlowJo LLC, Ashland, OR, USA). Cell multiplets, DTo ensure high viability and to exclude lysis-resistant nucleated RBCs found in neonates, 7-AAD+ dead cells, and CD235a+ erythrocytes were detected and excluded from FACS analysis. From viable, RBC-depleted cells, mixed immune cells (MICs) were identified by CD45.

### 2.7. Estimation of Maturation State of Human Cells (In Terms of Murine Maturity)

For the human cells, we estimated their individual developmental state in terms of post-natal days (PND) age of mouse lung cells. In order to look at different cell types independently, we isolated the human and mouse cells of same type from the integrated human and mouse cell Seurat object, and created new objects of each of those cell types. As a first step, within the Seurat object of each individual cell types, we applied Principal Component Analysis (PCA) as a means of dimension reduction. Once we identified the loadings for each of the cells for first ten principal components, we correlated the ages of the cells with the PC loadings, restricted to the mouse cells only. Based on the r values, we identified the PC with highest significant correlation with age and termed that as the age-associated PC or PC^Age^. 

Now, within this PC^Age^ space, for each human cell, we calculated the absolute distance between that human cell and every mouse cells of the same type. We subsequently identified the 100 nearest neighboring mouse cells from every human cell, in terms of sample loadings. We further adjusted the estimated age based on the proportion of mouse cells belonging to each of the time points (post-natal days 1, 3, 7, and 10). The estimated maturation stages of the individual human cells was calculated according to Equation (1) listed below.
*Estimated Maturity of Human Cells = Average {(Frequency × Age × Weight)_PND1_* +
*(In Terms of Murine Maturation)*       *(Frequency × Age × Weight)_PND3_* +                      *(Frequency × Age × Weight)_PND7_* +                      *(Frequency × Age × Weight)_PND10_}*(1)

### 2.8. In Situ Hybridization and Immunostaining

Fluorescence in situ hybridization (FISH), combined with immunofluorescence staining, was performed on formalin fixed, paraffin embedded native human postnatal lung sections (6 µm). FISH was completed using the RNAscope Fluorescent Multiplex Assay (Advanced Cell Diagnostics, Newark, CA, USA, cat. # 323110) as previously described [26,48], with minor adjustments. Treatment time with Protease Plus was reduced to 22 min. Tissues were incubated with the following probes: *HES1* (cat. # 311191-C4), *TCF21* (cat. # 470371) or *COL6A3* (cat. # 482631) (Advanced Cell Diagnostics). Following washing and signal development, tissues were blocked (3% bovine serum albumin in 5% normal goat sera and 0.1% Triton) and incubated overnight at 4 °C with primary antibodies: CD31 (Neomarkers, RB-10333-P0) or CDH1 (BD Biosciences, 6315829). Slides were washed and incubated with Cy3-goat-anti-mouse or anti-rabbit-conjugated secondary antibodies (Jackson Immunoresearch Laboratories, Inc., West Grove, PA, USA). Slides were counter-stained with DAPI (LifeTechnologies, Carlsbad, CA, USA, cat. # DE571) and mounted using ProLong Diamond Antifade Mountant (LifeTechnologies). Images were acquired on an LSM710 confocal system with a 20×/0.8 Plan-APOCHROMAT objective lens [48]. 

### 2.9. Statistics

Statistical analysis and calculations involving estimation of maturity state of human cells and additional analyses were performed Minitab 17 (Minitab Inc., State College, PA, USA). Additional images were generated using Graphpad Prism 6 and Microsoft Excel 2016. 

## 3. Results

### 3.1. Cellular Landscape of the Donor Newborn Human Lung

To characterize cellular heterogeneity in the newborn human lung, we performed single-cell RNA sequencing (scRNAseq) of protease-dissociated cells from two one-day-old donor lung samples (Donor 1: GAB 38 weeks, and Donor 2: GAB 31 weeks). While these donors were deceased on day one of their life due to non-lung related complications (anencephaly), the lungs appeared normal in appearance in histopathological analyses (Appendix A), and were processed with warm ischemic time. To exclude low quality events, cells having fewer than 500 genes detected, or with ≥12.5% mitochondrial genes, were excluded (Appendix A). Prior studies have shown that high levels of mitochondrial gene expression is an indication of presence of multiplets [49], and hence by using the filtering threshold of greater than 12.5% mitochondrial genes would exclude potential doublets from the analytical dataset. This filtering resulted in an analytical dataset of 19,136 genes in 5499 cells (Appendix A); 3001 cells from two separate captures on lung 1 and 2498 cells from two separate captures on lung 2. Since the two lungs were obtained from individuals born at different gestational age at birth (GAB), cells from each of the lungs were initially analyzed independently and their cellular composition was assessed at both major cell type level, as well as the level of cellular subtypes (Appendix A). Irrespective of the differences in terms of the estimated gestational age at birth, there were no observable differences in cellular composition of the two lungs, and hence cells from both lungs were combined using Canonical Correlation Analysis (CCA) as implemented in Seurat [43]. 

This data set was used for analysis and visualization by t-Distributed Stochastic Neighbor Embedding (t-SNE). We identified 15 separate clusters of cells, along with corresponding marker genes (Figure 1). Each cluster displayed relatively equivalent distribution of cells from both subjects (Figure 1a). Among these 15 clusters, four major cell types were identified on basis of expression of known markers (Figure 1b). Epithelial cells (*n* = 209) were defined by expression of *EPCAM*, *SFTPB*, *SCGB1A1*, and *NKX2-1.* Endothelial cells (*n* = 1092) were defined by expression of *PECAM1*, *VWF*, *CLDN5*, and *CDH5*). Mesenchymal cells (*n* = 3553) were defined by expression of *ACTA2*, *ELN*, *COL1A1*, and *CYR61*. Immune cells (*n* = 618) were defined by expression of leukocyte and lymphocyte cell markers *PTPRC*, *CD8A*, *CD19*, and *CD3E* (Figure 1c). 

Based upon selective expression, we identified marker genes for individual clusters (Figure 2a). Functional enrichment analysis successfully identified lung cell sub-types for each of the 15 clusters (Figure 2b and Appendix A). The markers for each individual cluster are presented in Appendix A.

A majority of the cells (>63%) appeared to be of mesenchymal origin. Distinct large populations of myofibroblasts (Cluster 0, *n* = 820), matrix fibroblasts (Cluster 1, *n* = 814), and smooth muscle cells (Cluster 2, *n* = 592) were identified. In addition, two distinct populations of pericytes were observed (Cluster 5, *n* = 419 and Cluster 6, *n* = 398). 

We also identified four separate endothelial cells clusters; Cluster 3 (*n* = 567), Cluster 8 (*n* = 321), Cluster 10 (*n* = 146), and Cluster 12 (*n* = 85). Interestingly, pathway analysis performed independently using the cluster marker genes using ToppGene Functional Annotation tool (ToppFun) [45], associated Cluster 3 cells with vascular development and Cluster 12 cells with integrin signaling pathways. 

A much smaller fraction of cells (<4%) were identified as epithelial cells. Epithelial cells were separated into AT1 cells (Cluster 11, *n* = 131) and AT2 cells (Cluster 14, *n* = 14). 

Interestingly, immune cells represented a sizeable fraction (11%) within the newborn human lung cells obtained from the two donors. Among immune cells (*n* = 649), we were able to distinguish multiple discrete populations including macrophages (Cluster 7, *n* = 349), T cells (Cluster 9, *n* = 190), and B cells (Cluster 13, *n* = 79). 

When analyzed independently, the proportion of immune cells varied between the two donors ranging from 17% in Donor 1 (GAB 38 weeks), to 4% in Donor 2 (GAB 31 weeks). While there have been prior reports of presence of immune cells in neonatal lungs, [50], we have previously shown the proportion of mixed immune cells in newborn lungs to range from 4 to 15% [39]. Further validation of the presence of these immune cells in newborn human lungs was performed by flow cytometry of single-cell dissociates from additional age-matched lungs (one day old) as described in the section on Flow Cytometry in the Appendix A. The percentage of leukocytes detected varied from donor to donor, and ranged from 3 to 14%, which was similar to the observed frequency of immune cells from the donor lungs used for generating the single-cell transcriptomics data set (Appendix A).

### 3.2. Cellular Landscape of the Postnatal Mouse Lung

Single-cell RNA sequencing of murine lung tissue was performed using custom Drop-seq technology as previously described [27]. Cells with fewer than 500 detected genes and greater than 12.5% of transcript counts mapped to mitochondrial genes were removed. Filtering resulted in an analytical data set of 17,508 genes, from 32,849 cells (PND1 *n* = 8003, PND3 *n* = 8090, PND7 *n* = 6324 and PND10 *n* = 10,432). All mouse cells were grouped into 32 clusters, and each cluster had relatively similar distribution of cells from individual time points (Figure 3a). Similar to the human cells, the four major cell types were readily identified based upon the expression of known selective marker gene expression (Appendix A). Mesenchymal cells, again represented the largest fraction of the population (*n* = 10,678; 33%) and were further identified into different sub-types including of multiple clusters of matrix fibroblasts, myofibroblasts, stromal cells, and mixed fibroblasts. Endothelial cells comprised a sizeable portion of the mouse cells (*n* = 8891), and compared to human newborn lungs, mouse lungs appear to have relatively greater proportion of epithelial cells (*n* = 6133), which were further sub-classified into pulmonary alveolar type I (AT1), alveolar type II (AT2), and ciliated respiratory epithelial cells. As in the human, immune cells were detected in the neonatal mouse lung (*n* = 7244) and were further classified as B-cells, T-cells, macrophages, monocytes, and myeloid cells among others (Figure 3b). 

### 3.3. Integration of Newborn Human and Mouse Lung Data Sets 

The two donor newborn human lungs differed in terms of gestational age at birth leading to minor morphological differences, however the cellular compositions were very similar across the two lungs (as shown in Appendix A) when it comes cellular annotations associated with the clusters of cells from individual donors. Admittedly, there would be differences in maturity status in the two donor lungs, however due to limited sample size, it is difficult to determine whether the differences between the two are due to technical variation (batch effect) or biological variation (maturity). In order to expand our analytical horizons, we subsequently combined the human and murine lung data sets (Figure 4). A total of 14,502 orthologous genes were identified using BioMart [46]. Canonical correlation analysis (CCA), implemented in Seurat, was used for data integration across the species. The combined dataset was filtered using the same criteria as applied independently on the human and mouse datasets (excluding cells having fewer than 500 genes detected, or with ≥12.5% mitochondrial genes), the species-integrated analytical data set contained a total of 29,762 cells: 2327 (15%) human cells and 27,435 (85%) mouse cells. The loss of some of the human and mouse cells can be attributed to the use of a limited set of genes (14,502) that were used for integrating the data across species, and thereby leading to those cells being filtered due to not passing the filtering thresholds. In this integrated data set, we identified 17 clusters of cells, along with corresponding cluster marker genes. Each cluster was composed of a combination of mouse and human cells (Figure 4a, Appendix A). We again identified four major cell types by known cell-type selective marker expression (Figure 4b–d); mesenchymal cells (*n* = 9980, 15% human), endothelial cells (*n* = 8292, 8% human), epithelial cells (*n* = 5146, 0.5% human) and immune cells (*n* = 3342, 6% human). 

Based upon selective expression, we identified marker genes for individual clusters (Figure 5a). Functional enrichment analysis successfully identified lung cell sub-types for each of the 17 clusters in the integrated data set representing postnatal lung tissues from human and mouse (Figure 5b and Appendix A). We observed multiple clusters of mesenchymal cells; myofibroblasts (*n* = 4592; Clusters 3, 10, and 11), matrix fibroblasts (*n* = 4743; Clusters 1 and 14) and pericytes (*n* = 645, Cluster 12). We observed multiple clusters of endothelial cells (*n* = 8292; Clusters 0, 5, 9, and 15). We observed multiple immune cell populations including macrophages (*n* = 3002, Cluster 4), T cells (*n* = 1330, Cluster 8), B cells (*n* = 1388, Cluster 7), and myeloid cells (*n* = 585, Cluster 13) as well. 

### 3.4. Estimating Developmental State of Human Cells

Lung development and morphogenesis occurs both prenatally and postnatally, and is typically divided into five phases, with the final alveolar phase occurring principally after birth in humans and rodents. The final stage of lung development, termed alveolarization or alveogenesis, begins prior to birth in humans and extends through at least the first decade of life, while occurring entirely postnatally in mice [18]. There exists a degree of uncertainty regarding the state of development states of the human and mouse lungs at the time of birth, since rodents and humans are born at different histological stages, alveolar in humans, but saccular in mice. The process of lung morphogenesis is a continuous process which varies by cellular types, as shown in newborn mouse lung [27]. Even though the two donor human lungs were collected at day 1 of birth, we hypothesized that due to the continuous nature of cellular development and dissociation, individual cells would be at different stages of their development throughout the process of lung morphogenesis. We sought to determine the “cellular developmental state” of the newborn human lung in comparison to the postnatal mouse lung, using the integrated human-mouse data set. Since a one-day old human lung is developmentally and morphologically similar to 5–7-day old mouse lung, we aimed to estimate the developmental stage or maturity of the human cells in terms of the corresponding murine maturation state by using the full spectrum of the available murine lung development data. Even if we consider the early gestational age of one of the donor lungs, the timespan covered by the mouse lung data complements the developmental state of the human lungs, and is therefore most appropriate background for integration and estimation of developmental state of the human cells. We performed this analysis separately for each distinct cell type independently (Appendix A), performed Principal Component Analysis (PCA), and tested the relationship between each PC vector and the known age of the mouse cells. We calculated an “estimated age” for every human cell using its linear distance to 100 mouse cells in space defined by the age-related PC (PC^Age^) that was statistically correlated with age and explained the greatest variance in the data set. The PC associated with age differed for each cell type, and the correlation coefficients (r) values, which were used as metric for identifying the PC related to age (Appendix A). Irrespective of the cell type, the largest influence on the variance within each of them was due to species differences, as demonstrated by cells separating by species on the first principal component (PC1). 

The estimated maturity of individual human cell types differed slightly, but primarily remained in the range of 5–9 murine days, consistent with the known histological relationships between human and mouse (Figure 6a). Interestingly, epithelial cells, endothelial cells and matrix fibroblasts displayed a more diverse distribution in estimated maturity. Matrix fibroblasts, which represented a large proportion of all cells displayed a somewhat bi-phasic pattern, where 29% of cells appeared to be an estimated maturity stage consistent with other cell types (5–9 days), while a second set of cells appeared to be of much younger estimated maturity stage (1–4 days) (Figure 6b and Appendix A). Even though the maturation stages have been defined in terms of murine lung cell maturity, it is significant that we observe two different groups of cells that have been classified as matrix fibroblast. Both the early and late maturity matrix fibroblasts were evenly distributed across both samples and showed no statistical difference in distribution across the samples. We identified marker genes for younger and more mature matrix fibroblast population using DESeq2 [51] leading to identification of 210 differentially expressed genes, of which 23 genes were over expressed in “the older matrix fibroblasts” (those presenting with an older estimated age), while 187 genes were over expressed in “younger matrix fibroblasts” (those presenting with a younger estimated age). When queried for cell types associated with these genes, we mostly observed then to be associated with lung or matrix fibroblasts, however, a subset of 44 (out of 187) genes were found to be associated with progenitor fibroblasts. Pathway analysis using these 187 genes revealed smooth muscle, matrix, and collagen related pathways, along with oxidative stress, stress response, degranulation and scavenging related pathways were upregulated in the immature matrix fibroblasts (Appendix A).

One of the markers for younger matrix fibroblasts which had over two-fold induction was *HES1*, which was also expressed in other mesenchymal cells (pericytes and stromal cells) and endothelial cells as well (Appendix A). We identified human cells expressing *HES1* alone, and cells co-expressing *HES1* with matrix fibroblast markers (*COL6A3* or *TCF21*; Appendix A) or markers for other cell types (*PECAM1* for endothelial cells or *CDH1* for epithelial cells; Appendix A). We further demarcated both human and mouse cells expressing HES1 alone (Appendix A, and cells co-expressing *HES1* with matrix fibroblast markers (*COL6A3* or *TCF21*; Appendix A) or markers for other cell types (*PECAM1* for endothelial cells or *CDH1* for epithelial cells; Appendix A). In both instances, we observed *HES1* being co-expressed mesenchymal markers in human cells in cluster(s) identified as matrix fibroblasts. Finally, we tested whether we could spatially resolve the older and younger matrix fibroblasts in the newborn human lung. We performed combined immunohistochemistry and in situ hybridization in three independent donor lungs of same age (one day old), to identify the expression of general- (*COL6A3* and *TCF21*) and immature population-specific (*HES1*) markers at the cellular level. We were able to identify individual matrix fibroblasts (as defined by expression of *COL6A3* or *TCF21*, but not *PECAM1* or *CDH1*) that expressed HES1, as well as matrix fibroblasts that did not express *HES1* (Figure 6c and Appendix A). These data indicate the presence of a distinct group of immature matrix fibroblasts in the newborn human lung that display high expression of *HES1* transcript. 

## 4. Discussion

By applying single-cell RNA sequencing to newborn human lungs, we identified a diversity of pulmonary cells, including epithelial, fibroblast, immune, endothelial, and other cell subtypes based upon distinct gene expression patterns. We note a paucity in the capture of epithelial cells, which is different from previous reports [32,33,34], but is consistent with our recent report of similar analyses of human fetal lung tissues [26]. Our prior studies using similar cell isolation protocols in older pediatric lung samples, demonstrated a higher proportion of epithelial cells [39]. Similar to our prior studies applying similar methodologies, we observed an over-representation of mesenchymal cells, which were further classified into subtypes, namely fibroblasts, pericytes, and stromal cells [26]. We also observed a sizeable cluster of endothelial cells, however, the number of cells were not sufficient to identify rare capillary-type endothelial cell types seem in mouse lung cells [52]. Our analysis also indicates the presence of immune cells in the newborn lungs, which has not been widely reported before. While there have been multiple recent studies involving single-cell sequencing of the human lung [32,35,36,53], this is one of the first studies reporting the cellular composition of human lungs at the time of birth. It has recently been shown in the adult lung cells that NK cells tend to be relatively close in spatial resolution to CD8 T cells [54]. Resident memory T cells are reported to be the most abundant T cells detected in peripheral blood, however they present in multiple in tissues under stable conditions [55]. A recent review article highlights the fact pediatric T cells often exhibit a more effector-like phenotype and therefore resemble innate-like immune cell populations, it is possible that resolution of T cells and NK cells is difficult [56]. Given that the lungs were recovered from one-day-old donors, and the presence of maternal immune cells has been reported, it is possible that some of the immune cells are of maternal origin [57]. Recent transcriptomic studies using single-cell sequencing from adult human lung tissues have revealed existence of the major cellular lineages, albeit with variations in proportion [32,58]. Recently, the Human Lung Cell Atlas (HLCA) has been developed by integrating multiple single-cell sequencing datasets from human lung samples [54]. The integrated HLCA demonstrated greater proportion of cellular lineages of epithelial and immune origin, and a reduction in proportion of mesenchymal and endothelial lineages, compared to our observations in the newborn lungs. However, the HLCA also demonstrated that the different fibroblast subtypes are associated with inherited differences in lung functions, which correlates with our findings of different maturation rates in the matrix fibroblasts [54]. Additionally, reports in mice have indicated that the presence of naïve immune cells could be located in the vasculature of lung and not in the tissue [59], although our recovery protocol includes a flush step which will help to reduce the presence of immune cells in the vasculature of the lung. Further immuno-staining studies will be necessary to identify the spatial location of cells from the adaptive immune system in one-day-old donors.

Although this is a novel and necessary study, it has to be acknowledged that some bias likely exists in the cells isolated and described. While the sensitivity of single-cell RNA sequencing allows the discovery of high-resolution cellular topography, it also gives rise to susceptibility to technical and procedural biases which sometimes hide the true biological signals [60]. Cellular recovery can be impacted by the methods used for mechanical and enzymatic disaggregation, processing time and reagent concentrations. Developing enzyme-mediated digestion conditions will depend on which assays will be completed downstream. Our group’s initial goal was to optimally release all cell types from primary human lung, including fibroblasts which are held in place by the proteinaceous extracellular matrix [39,61]. The process requires careful testing of how changing conditions affects the quality of sample generated [62]. While aggressive or prolonged digestion protocols may lead to cell death or fragmentation [63], gentle dissociation protocols may lead to greater capture of easily dissociated cells, which might be a challenge in the lung diseases characterized by altered matrix structures [64]. We did observe an absence in type II epithelial cell capture prior to selection, suggesting epithelial cell viability may contribute their diminished detection. In fact, in a review of published datasets, Alexander et al., have reported that in normal lung tissue cell types, such as broad, flat, sail-like alveolar type I (AT1) cells and matrix-embedded fibroblasts, they are difficult to dissociate and liberate and as such are relatively underrepresented when compared with cuboidal alveolar type II (AT2) cells and alveolar macrophages [63]. Even within the limited number of epithelial cells, we observed the expression of markers (*SFTPC*, *AQP5*, *HOPX*) of alveolar epithelial differentiation in alveolar type II (AT2) cells [65] obtained from newborn lungs. Low capture/detection of epithelial cells in the fetal and newborn human lung may be attributed to the developmental age of the studied samples, cellular stress encountered due to the dissociation protocols or difficulty in capture of these cells with the Chromium 10X protocol. Cellular capture from the young lungs is further complicated by the lack of knowledge regarding cell dissociation, different protease sensitivity, and cell survival during digestion and capture procedures. We performed two independent captures on each sample, one involving selection for high-quality cells by removing dead or dying cells by magnetic selection. Importantly, we noted consistent recovery of all major cell populations regardless of capture (Appendix A). Furthermore, the use of dissociated cells has been used by our group to develop an in vitro air liquid interface cell culture model system [6,9]. This provides an opportunity for further refinement of dissociation protocols to improve the recovery of epithelial cell populations [26]. 

It is clear that although stages of lung development, and their morphological correlates, are highly conserved across species, significant differences exist in their relative length and timing [48]. An example is that the mouse lung is in the saccular stage at birth, while the human lung at term birth is in the alveolar stage. The newborn human lung is histologically and developmentally similar to a one-week-old mouse lung (Appendix A). We took advantage of recent data from the LungMAP program, describing postnatal mouse lung development at the single-cell level, to infer the “estimated maturity” or “developmental state” of newborn human lung cells [27]. 

We have analyzed and annotated the species-specific single-cell datasets separately which helped us identify major cell types, as well as cellular subtypes. While the separate analyses preserve the intra-dataset heterogeneity, the combined analyses across the two species (human and mouse) increases the number of cells used for clustering and annotation, potentially allowing for identification of additional heterogeneity and rare cell populations. However, the combinatorial analytical process is more complex and computationally intensive, and may undermine some of the species-specific cell types [66]. In the combined analyses, species-level differences can be minimized by applying “batch-correction” approaches on the underlying transcriptomic data, and we have incorporated that by using gene names matched across species [46], and performing subsequent integration using CCA [43]. In order to alleviate concerns regarding the retention of the lineage and cell type information of the human cells when integrating with the murine cells, we tested for the cell type identities and observed that over 96% of human cells identified as mesenchymal, endothelial, and immune cells retain their identities in the combined classification as well (Appendix A). Even in case of epithelial cells, which were only 24 to begin with, we observed 75% accuracy. 

Our single-cell data was generated from two donated lungs that were of one day of age, one of which was born full time (at 38 weeks of gestation), while the other was born prematurely (at 31 weeks of gestation). The lung as we know is a heterogeneous organ and is composed of over 40 cell types. We hypothesized that different cell types may have different stages of development during the process of development. In order to assess the differential growth of developmental states of different cell types, we have integrated the human cells with the single-cell data from mouse lungs harvested from animals at different ages of development (post-natal days 1, 3, 7, and 10).

The majority of the human cells, regardless of cell type/lineage, were estimated to be 4 to 9 days of mouse age, consistent with the histological comparisons. For some cell types (e.g., matrix fibroblasts, endothelial cells, epithelial cells), greater diversity in estimated age was noted. The epithelial cell population was not large enough to separate known distinct lineages. Among the other cellular populations, the extent of endothelial cell diversity (e.g., large vs. small vessel), has been well-documented [67]. We focused subsequent analysis on the matrix fibroblasts, as phenotypic diversity among this population is less well-described. 

Our observations on newborn human lung matrix fibroblast diversity are possibly one of the first to be reported in humans, but there has been a prior report of different types of murine lung matrix fibroblasts in a mouse model of pulmonary fibrosis [68]. While Xie et al. [68] focused on mouse lung fibroblasts, they did observe a class of murine progenitor lung fibroblasts. Interestingly, a set of 44 (out of 187) genes unregulated in the early matrix fibroblasts in our newborn human lungs were found to be associated with progenitor lung fibroblasts. The majority of newborn human lung matrix fibroblasts appeared to be more similar to younger mouse matrix fibroblasts and displayed higher levels of expression of *HES1*. These less-mature fibroblasts were evenly derived from both the newborn lungs, as there was no significance difference in distribution observed across the two donor lungs irrespective of the difference in gestational ages of the two. *HES1* is a regulator of Notch signaling and appears to actively suppress differentiation [69]. Interestingly, the regulation of collagen expression by Notch is achieved through a HES1-dependent mechanism [70]. Furthermore, *HES1* appears to play a critical role in regulating lung fibroblast differentiation [71], and is known to be expressed in mucus cells from patients with chronic obstructive pulmonary disease, idiopathic pulmonary artery hypertension or IPF [72]. Hes1-knockout mice have been reported to have a phenotype of premature differentiation and rigorous defects in nervous tissue [73]. Another gene that displayed higher expression in the younger/immature matrix fibroblasts was *IGFBP7*, which has previously been associated with resistance to lung cancer by performing tumor suppression function, especially in epithelial cells [74]. The younger/immature matrix fibroblasts may represent cells in an immature state, with importance for normal development, and may hint at a developmental origin for some adult diseases such as lung fibrosis.

We understand that the study is limited to the data from only two one-day-old human lung samples, which are difficult to obtain. While these samples may not be the ideal healthy lung samples due to them being obtained from deceased subjects, who have developmental anomalies in other organs, however, we would like to reiterate that we have used utmost caution to select the lungs samples to be user that are histologically normal (Appendix A), and controlled for ischemic time. Also, while the two samples differ in their gestational age, independent analysis of the two samples indicated similar cellular composition (Appendix A), thereby alleviating any concerns regarding age specific differences. Additionally, there is an underrepresentation of epithelial cells among the two subjects, which can be attributed to the dissociation protocol, which has previously been documented [39], and hence leaves the room for further refinement of the protocols leading to improved recovery of the pulmonary epithelial cells. However, when integrated with the mouse lung cells, they provide sufficient information to develop a molecular map of the human neonatal lung, and potentially overcome the limitation of low sample numbers.

To summarize, here we report a dataset describing the transcriptome of newborn human lung cells defined using single-cell RNA sequencing. Our results include markers for all major lung cell types including multiple populations of mesenchymal, endothelial, epithelial and immune cells. These lineage markers have been validated in the major pulmonary cells types through bulk RNA-seq [39]. We also successfully integrated the transcriptomes of newborn human cells with postnatal developing mouse lung cells, enabling the estimation of cell-type specific developmental states of human cells. The data show that maturation states, even though largely in the expected range of 4 to 9 murine postnatal days, differ by cell type. Integrated single-cell RNA profiling of human and mouse lung will help identify common and species-specific mechanisms of lung development and respiratory disease. Even with the limited sample size, our novel observations can provide valuable insights on the mechanisms of normal lung development.

## Figures and Tables

**Figure 1 genes-15-00298-f001:**
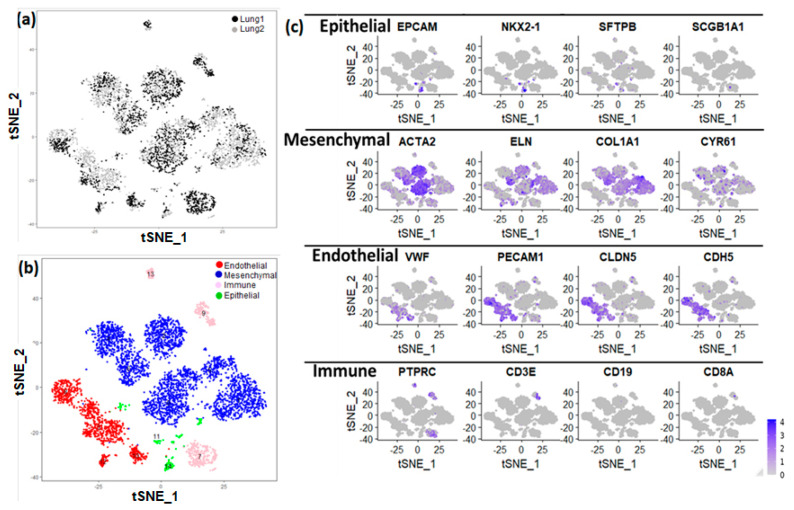
Identification of lung major cell types using single−cell RNA sequencing of newborn human lung. (**a**) t−Distributed Stochastic Neighbor Embedding (tSNE) analysis of cells. Cells are indicated by donor (lung 1: GAB 39 weeks; lung 2: GAB 31 weeks; GAB: Gestational Age at Birth). (**b**) The assignment of cell clusters to four major cell types, including endothelial cells, mesenchymal cells, immune cells, and epithelial cells. (**c**) Expression of some known cell type markers.

**Figure 2 genes-15-00298-f002:**
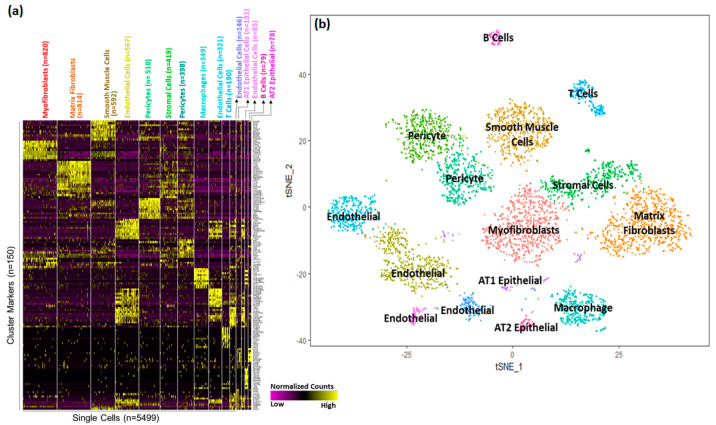
Identification of cell sub−type markers in newborn human lungs. (**a**) Gene expression patterns of select markers for corresponding cell clusters (5499 cells obtained from two newborn human lungs. (**b**) Assignment of cell types to 15 distinct cell clusters.

**Figure 3 genes-15-00298-f003:**
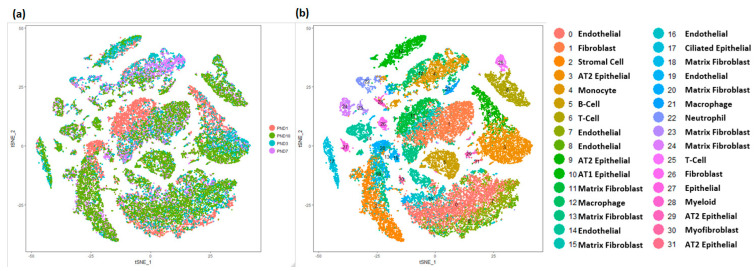
Identification of cell types in mouse lung. (**a**) t−Distributed Stochastic Neighbor Embedding (tSNE) analysis of cells. Cells are colored by mouse age. (**b**) Visualization of cell clusters in tSNE plot of cells, with assignment of cell types to 32 distinct tSNE clusters. Mouse lung transcriptomic profile data was generated at Cincinnati Children’s Hospital and Medical Center (CCHMC) [27].

**Figure 4 genes-15-00298-f004:**
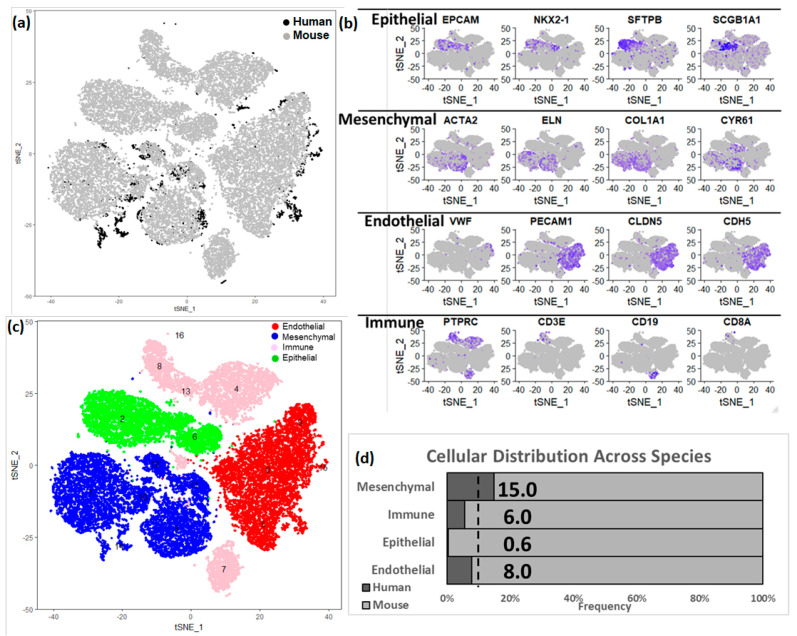
Integration of human and mouse lung cell data sets. (**a**) t−Distributed Stochastic Neighbor Embedding (tSNE) analysis of cells. Cells are indicated by species. (**b**) Expression of known cell type markers in tSNE plot of cells in the integrated data set. (**c**) Visualization of mouse cell clusters in tSNE plot of cells grouped by major cell types. In the integrated object created from combining both human and mouse lung cells, the assignment of cell clusters to four major cell types, including endothelial cells, mesenchymal cells, immune cells, and epithelial cells. (**d**) Proportion of cells derived from human and mouse data. In the integrated dataset 9% of cells are human (as indicated by the dotted line); human mesenchymal cells (15%) are over-represented, but endothelial (8.0%), immune (6.0%) and epithelial cells (0.5%) are under-represented.

**Figure 5 genes-15-00298-f005:**
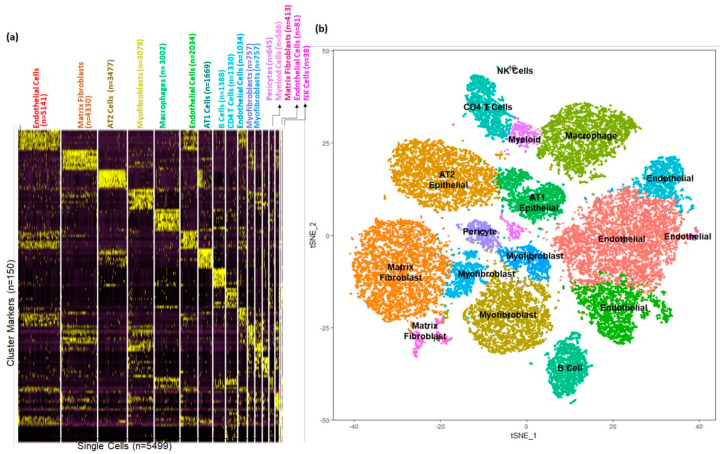
Identification of cellular sub−types in combined human and mouse lungs. (**a**) Gene expression patterns of select markers for corresponding cell types is shown in the heatmap. (**b**) The *t*-Distributed Stochastic Neighbor Embedding (tSNE) visualization shows unsupervised transcriptomic clustering, revealing 17 distinct cellular identities.

**Figure 6 genes-15-00298-f006:**
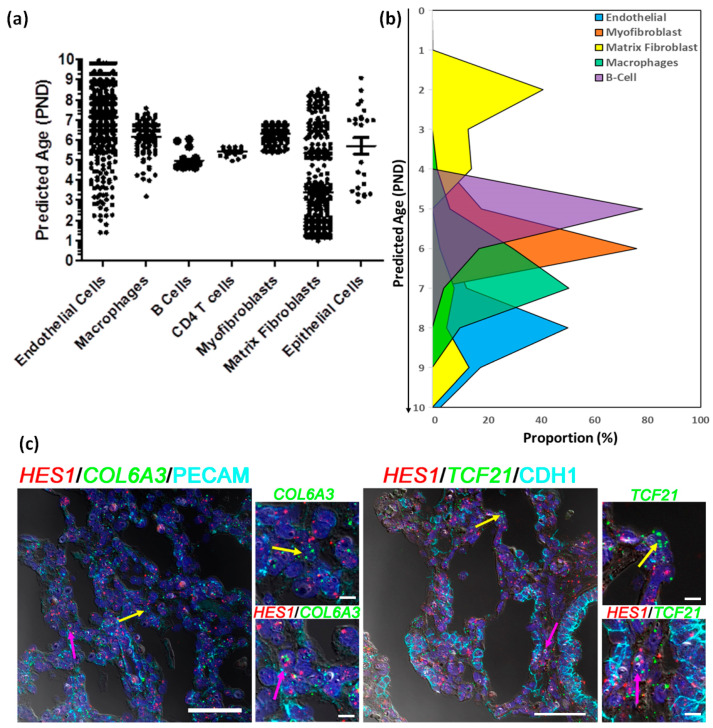
Estimating developmental states of human cells. (**a**) Distribution of the estimated ages of the human cells derived from post -natal age (PND) of 100 nearest mouse cells to each of the human cells. (**b**) Proportion of cells of individual human cell types at each stage of development defined in terms of estimated post-natal day age of mouse. (**c**) Fluorescent in situ hybridization (FISH) combined with immunofluorescence of Immature Matrix Fibroblast marker *HES1* (red), Non-Mesenchymal Cell Markers, *PECAM1* or *CDH1* cyan), and Mesenchymal Cell Markers *COL6A3* or *TCF21* (green) on newborn human lung sections from a donor lung of 1 day of age. Pink arrows indicated the presence of immature matrix fibroblasts shown by co-localization of *HES1* (red) and Mesenchymal Cell Markers *COL6A3* or *TCF21* (green), while yellow arrows indicate non-*HES1* expressing cells. The scale bar is 50 µm. PND: Post-Natal Days.

## Data Availability

The human cell sequencing counts data is available for visualization and downloading through the LungMAP portal (https://lungmap.net/breath-omics-experiment-page/?experimentTypeId=LMXT0000000016&experimentId=LMEX0000004390&analysisId=LMAN0000000347&view=entitySet) last accessed on 20 February 2024. The mouser data from single-cell RNA-seq experiments have been deposited in Gene Expression Omnibus under accession code [GSE122332].

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
