# Peer review of "Single-Cell Transcriptomic Profiling Identifies Molecular Phenotypes of Newborn Human Lung Cells"

_genes, 2024, doi:10.3390/genes15030298_

Round 1
Reviewer 1 Report
Comments and Suggestions for Authors
The manuscript of Bhattacharya et al. presents the results of the single-cell transcriptomics performed on two human neonatal samples and murine samples. While data on prenatal human lung have already been created, those describing the newborn human lung are lacking. The authors performed sc RNA Seq on samples from two human donors with anencephaly 1 day postnatally. One of them was born at 38, the other at 31 weeks. Human and mice data have been integrated and mice data has been used as a reference. The marker genes and clusters have been identified. The majority of the cells were of mesenchymal origin. Interestingly there was a significant fraction of immune cells, however varied between the donors. Also, in mice 4 major types of cells have been identified. The maturity of the cells in comparison to murine data has been estimated. The figures are of tge good quality and illustrative. Extrensive data are addedd as a supplement. In the discussion the authors compare the available data with the data from older donors and mice and acknowledge potential bias due to the technical issues and the age/number of samples in donors. This is an important contribution to the current state of knowledge, expanding on the cellular composotion of the human lungs in newborns.
Minor comments:
Line 52 "will be required further" please doble check this sentence gramatically
Line 64 Please describe shortly what the NHLBI Molecular Atlas of Lung Development Program (LungMAP) is. Is it open source? Please provide the website address
Line 69-70 This sentence is wordy
Line 97-Did the donors undergo standard genetic testing to exclude genetic background? Was the reason for anencephaly known`?
Please write human genes in italics thoughout the text, also in the supplement
Discussion: Please add shortly how these findings relate to the data from adult human lungs.
Author Response
Genes-2877606 Response to comments from Reviewer 1
The manuscript of Bhattacharya et al. presents the results of the single-cell transcriptomics performed on two human neonatal samples and murine samples. While data on prenatal human lung have already been created, those describing the newborn human lung are lacking. The authors performed sc RNA Seq on samples from two human donors with anencephaly 1 day postnatally. One of them was born at 38, the other at 31 weeks. Human and mice data have been integrated and mice data has been used as a reference. The marker genes and clusters have been identified. The majority of the cells were of mesenchymal origin. Interestingly there was a significant fraction of immune cells, however varied between the donors. Also, in mice 4 major types of cells have been identified. The maturity of the cells in comparison to murine data has been estimated. The figures are of tge good quality and illustrative. Extrensive data are addedd as a supplement. In the discussion the authors compare the available data with the data from older donors and mice and acknowledge potential bias due to the technical issues and the age/number of samples in donors. This is an important contribution to the current state of knowledge, expanding on the cellular composotion of the human lungs in newborns.
We thank the reviewer for their helpful and encouraging comments.
Minor comments:
Line 52 "will be required further" please doble check this sentence grammatically
We apologize for the typographical error and have fixed the statement.
Line 64 Please describe shortly what the NHLBI Molecular Atlas of Lung Development Program (LungMAP) is. Is it open source? Please provide the website address
We thank the reviewer for the suggestion, and have now provided additional information about LungMAP consortium. We have also provided the website for the LungMAP consortium in the introduction section.
Line 69-70 This sentence is wordy
We thank the reviewer for their suggestion, and have now broken the statement into two sentences which can now be better interpreted.
Line 97-Did the donors undergo standard genetic testing to exclude genetic background? Was the reason for anencephaly known`?
Since the lung tissue was obtained through donations through the federal United Network of Organ Sharing via the National Disease Research Interchange (NDRI) and International Institute for Advancement of Medicine (IIAM), we are only privy to the clinical information provided to us by the network. While we would like to know additional details about the donors, but unfortunately we do not know the reason for anencephaly.
Please write human genes in italics thoughout the text, also in the supplement
We have now updated text and tables in the manuscript and the supplement with italicized human gene names.
Discussion: Please add shortly how these findings relate to the data from adult human lungs.
We thank the reviewer for the suggestion. We have reviewed recent transcriptomic studies using single cell sequencing from adult human lung tissues which have revealed existence of the major cellular lineages, albeit with variations in proportion, compared to newborn lungs. Recently published integrated Human Lung Cell Atlas (HLCA) data has shown greater proportion of epithelial and immune cells in adult lungs, compared to newborn lungs. However HLCA did report association of fibroblast subtypes with lung function variability, which corresponds to our observation of variation in maturity rates within the different fibroblast types in the newborn lungs. We have included these observations in the discussion section.
Reviewer 2 Report
Comments and Suggestions for Authors
What is the main question addressed by the research?
Soumyaroop Bhattacharya et al. report a transcriptional analysis of human lung cells from two diagnosed anencephaly donor newborns, including all major cell types, and describe the molecular profile that may represent cells at different stages of development.
Do you consider the topic original or relevant in the field? Does it address a specific gap in the field?
Yes, it is an original and relevant topic, there is still a lack of information on the topic.
What does it add to the subject area compared with other published material?
The authors show us what cells and transcription of cells are at a very early stage of transcription at a very early stage of human development
Are the conclusions consistent with the evidence and arguments presented, and do they address the main question posed? Yes, they are
Are the references appropriate? Yes, they are
What specific improvements should the authors consider regarding the methodology? What further controls should be considered? include any additional comments on the tables and figures.
Abstract: The authors are recommended to put the aim of the study implicitly.
I would like to see the normality of the lung tissues used to be shown on H and E staining (as in line 228).
I understand the difficulty of obtaining healthy lung samples in these days of development, however I think you should discuss the disadvantages of using the sample used in this analysis, as there will always be the question of "healthy lungs".
Author Response
Genes-2877606 Response to comments from Reviewer 2
Abstract: The authors are recommended to put the aim of the study implicitly.
We thank the reviewer for the suggestion. We have added the statement ‘Transcriptomic profiling at the single cell level created a cellular atlas of newborn human lungs.’ In abstract implying the goal of the study.
I would like to see the normality of the lung tissues used to be shown on H and E staining (as in line 228).
We thank the reviewer for the suggestion and have included H and E staining images for two donor lungs from the same lobes that were used for sequencing. As indicated in the manuscript, both the samples were histopathologically normal, and showed lung structure according to their gestational age. H&E staining images of histological sections is presented in Figure S10.
I understand the difficulty of obtaining healthy lung samples in these days of development, however I think you should discuss the disadvantages of using the sample used in this analysis, as there will always be the question of "healthy lungs".
We thank the reviewer for understanding the rare nature of these samples, and we would like to reassure that when selecting the samples that showed normal in appearance in histological analyses (added Figure S10) and were processed with warm ischemic time. We have expanded the limitations paragraph in the Discussion section to include text acknowledging that these are not the ideal healthy normal but are the best choices under the given conditions.
Reviewer 3 Report
Comments and Suggestions for Authors
While Bhattacharya et al. have tried to answer a very important question in the field, which is to define the transcriptional map of newborn human lungs and to compare it with mice, this manuscript has some major limitations:
1. They have used only 2 donor lungs. This sample size is very small, diminishing confidence in the findings.
2. It is a major problem that the 2 donor lungs are from two different gestational ages – 31 and 38 weeks. This is a very critical time for lung development, and it does not seem reasonable to combine these two time points.
3. For mouse data, only 2 samples at each time point were used – it is advisable to use at least 3 mice from each time point to be able to make statistically significant inferences about each time point.
4. All p values in table s2 are zero – what does that mean?
In summary, while this is an important endeavor, data robustness should be increased for confidant interpretations.
Author Response
Genes-2877606 Response to comments from Reviewer 3
While Bhattacharya et al. have tried to answer a very important question in the field, which is to define the transcriptional map of newborn human lungs and to compare it with mice, this manuscript has some major limitations:
We thank the reviewer for their thorough review and helpful comments. We have addressed their concerns below and responded individually to their concerns.
- They have used only 2 donor lungs. This sample size is very small, diminishing confidence in the findings.
We understand the reviewers concern that the study is limited to the data from only two one day old human lung samples, which are difficult to obtain. Even with the limited sample size, our novel observations can provide valuable insights on the mechanisms of normal lung development. We have added the statement in the discussion section.
- It is a major problem that the 2 donor lungs are from two different gestational ages – 31 and 38 weeks. This is a very critical time for lung development, and it does not seem reasonable to combine these two time points.
We appreciate the reviewers concern regarding differences in maturity states of the two donor human lungs. However, we would like to point out that the goal of the study was to characterize the cellular composition of the newborn human lung. While the two donor lungs differed in terms of gestational age at birth leading to expected morphological differences, however the cellular compositions were not identical, but were very similar (as shown in Figure S3) when it comes cellular annotations associated with the clusters of cells from individual donors. Admittedly, there would be differences in maturity status in the two donor lungs, however due to limited sample size, it is difficult to distinguish the differences between the two are due to technical variation (batch effect) or biological variation (maturity). That is why we have enhanced the data size by incorporating the mouse single cell dataset. In further analyses we observed that both the younger, and older matrix fibroblasts were evenly distributed across both samples and showed no statistical difference in distribution across the samples (Table S11).
- For mouse data, only 2 samples at each time point were used – it is advisable to use at least 3 mice from each time point to be able to make statistically significant inferences about each time point.
We would like to clarify that for the mouse data, at each time point, lungs were isolated and pooled from three newborn mice, and subsequently the left and right lobes were mixed and digested through enzymatic cocktail to isolate individual cells. The single-cell suspensions from each team point were then were processed through two independent Drop-Seq runs to generate two sets single-cell sequencing data at each of the four time points. The process is described in details in the original publication of Guo et. al., 2019 Nature Communications (PMID: 30604742).
- All p values in table s2 are zero – what does that mean?
The p-values presented in Table S2 are adjusted or corrected for multiple testing. The adjusted p-values were rounded to the nearest single decimal places, and hence any value less than 0.05 appeared as 0.0. We have now expanded adjusted p-values to second the decimal place in the table S2.
In summary, while this is an important endeavor, data robustness should be increased for confidant interpretations.
We understand that the study has its limitations due to small sample sizes. However we would like to reiterate that the study presents data from newborn human lung tissues, which are difficult to obtain. Even with the limited sample size, our novel observations can provide valuable insights on the mechanisms of normal lung development. We feel that these observations are relevant, and novel enough, to be disseminated to the research community. We further believe that our study being one of the earliest using newborn lung tissues will serve as preliminary data for subsequent studies and can potentially be cited widely by the researchers in the field.
Round 2
Reviewer 3 Report
Comments and Suggestions for Authors
Thank you